# High-sensitivity analysis of clonal hematopoiesis reveals increased clonal complexity of potential-driver mutations in severe COVID-19 patients

Chiara Ronchini[1]*, Chiara Caprioli[2], Gianleo Tunzi[1¤a], Francesco Furio D'Amico[1¤b], Emanuela Colombo[2,3], Marco Giani[4,5], Giuseppe Foti[4,5], Donatella Conconi[4], Marialuisa Lavitrano[4], Rita Passerini[6], Luca Pase[7], Silvio Capizzi[8], Fabrizio Mastrilli[8], Myriam Alcalay[2,3], Roberto Orecchia[9], Gioacchino Natoli[2], Pier Giuseppe Pelicci[2,3]*

1 Clinical Genomics, European Institute of Oncology IRCCS, Milan, Italy, 2 Department of Experimental Oncology, European Institute of Oncology IRCCS, Milan, Italy, 3 Department of Oncology and Hemato-Oncology, University of Milan, Milan, Italy, 4 School of Medicine and Surgery, University of Milano-Bicocca, Monza, Italy, 5 Department of Emergency and Intensive Care, Ospedale San Gerardo, Monza, Italy, 6 Division of Laboratory Medicine, European Institute of Oncology IRCCS, Milan, Italy, 7 Occupational Medicine, European Institute of Oncology IRCCS, Milan, Italy, 8 Medical Administration, CMO, IEO, European Institute of Oncology, IRCCS, Milan, Italy, 9 Scientific Directorate, European Institute of Oncology IRCCS, Milan, Italy

¤a Current address: Euroclone, Pero (MI), Italy
¤b Current address: Hematology Department, Ospedale Niguarda, Milan, Italy
* chiara.ronchini@ieo.it (CR); piergiuseppe.pelicci@ieo.it (PGP)

**Data Availability Statement:** All relevant data are within the paper and its Supporting information files.

## Abstract

Whether Clonal Hematopoiesis (CH) represents a risk factor for severity of the COVID-19 disease remains a controversial issue. We report the first high-sensitivity analysis of CH in COVID-19 patients (threshold of detection at 0.5% vs 1 or 2% in previous studies). We analyzed 24 patients admitted to ICU for COVID-19 (COV-ICU) and 19 controls, including healthy subjects and asymptomatic SARS-CoV2-positive individuals. Despite the significantly higher numbers of CH mutations identified (80% mutations with <2% variant allele frequency, VAF), we did not find significant differences between COV-ICU patients and controls in the prevalence of CH or in the numbers, VAF or functional categories of the mutated genes, suggesting that CH is not overrepresented in patients with COVID-19. However, when considering potential drivers CH mutations (CH-PD), COV-ICU patients showed higher clonal complexity, in terms of both mutation numbers and VAF, and enrichment of variants reported in myeloid neoplasms. However, we did not score an impact of increased CH-PD on patient survival or clinical parameters associated with inflammation. These data suggest that COVID-19 influence the clonal composition of the peripheral blood and call for further investigations addressing the potential long-term clinical impact of CH on people experiencing severe COVID-19. We acknowledge that it will indispensable to perform further studies on larger patient cohorts in order to validate and generalize our conclusions. Moreover, we performed CH analysis at a single time point. It will be necessary to consider longitudinal approaches with long periods of follow-up in order to assess if the COVID-19

**Funding:** The present work was partly funded by the Italian Ministry of Health, RF-2019-12370784 to MA. The funder had no role in study design, data collection and analysis, decision to publish, or preparation of the manuscript.

disease could have an impact on the evolution of CH and long-term consequences in patients that experienced severe COVID-19.

## Introduction

Several lines of evidence suggest a mechanistic link between COVID-19 disease and the pre-existing condition of Clonal Hematopoiesis (CH). Clinically, the presence of somatic mutations at a VAF $\geq$ 2% ($\geq$4% for X-linked gene mutations in males) in genes associated to myeloid malignancies in the blood or bone marrow of individuals without a diagnosed hematologic disorder and without unexplained cytopenia is currently defined has clonal hematopoiesis of indeterminate potential (CHIP) [1]. CH and CHIP are highly prevalent in the elderly (10–20% in >70 years old individuals) [2–5]. Mechanistically, CH results from spontaneous occurrence of aging-associated mutations in normal hematopoietic stem cells, selection of variants conferring growth advantage and progressive expansion of one or more cellular clones. Mutations in genes involved in epigenetic regulation such as *DNMT3A*, *TET2* and *ASXL1* account for the majority of CH cases [3–5].

CH is associated with increased blood cancer risk, with a rate of disease progression that is affected by the size of the clone, numbers and type of mutations [6]. Notably, CH has been also linked to increased risk of all-cause mortality and cardiovascular diseases (myocardial infarction, stroke, venous thrombosis, chronic obstructive pulmonary disease), with an associated hazard ratio equal or greater than common cardiovascular risk-factors such as smoking, cholesterol levels and hypertension [7, 8]. Emerging evidences suggest that the increased risk of cardiovascular disease is due to a stronger inflammatory response induced by the CH-associated mutations, suggesting that CH may be a general factor in age-related inflammation and disease [9–13]. Similarly, it has been recently reported an association between CHIP and presence of liver inflammation and fibrosis, that double the risk of CHIP carriers of developing chronic liver disease [14]. Studies in model systems suggest that mutant macrophages induce an inflammatory milieu which contributes to clonal expansion and associated comorbidities [14, 15].

COVID-19 prevalence and/or severity are associated with features that are similar to those observed in CH, including increased severity with age, with a lower percentage of survivors in the 63–76 years old range, and increased incidence of coagulation disorders (disseminated intravascular coagulation and venous thromboembolism) and cardiovascular diseases (cardiac ischemia). Notably, cardiac injury is common in COVID-19 hospitalized patients and COVID-19 is associated with abnormal activation of tissue macrophages and systemic hyperinflammation [16–21].

The impact of CH on the incidence and the severity of COVID-19 is still a debated issue, considering that different studies reported opposite results. The data from Bolton et al. [22], Duployez et al. [23], Li et al. [24] and Kang et al. [25] showed a higher prevalence of CH in COVID-19 patients and an increased risk for severe COVID-19 disease in presence of CH. In contrast, data from Hameister et al. [26], Petzer et al. [27] and Zhou et al. [28] showed that CH is not overrepresented in COVID-19 patients and its presence does not appear to be relevant for the severity of the disease.

The aim of our study was to analyze the presence and the impact of CH in COVID-19 patients using high sensitivity approaches enabling the identification of low-frequency clones.

## Materials and methods

### Patient cohort

Our cohort of patients includes hospitalized patients with confirmed SARS-CoV-2 infection by molecular testing and admitted to intensive care unit (ICU) of the San Gerardo Hospital, Monza Italy, because of severe pneumonia requiring invasive mechanical ventilation. Samples were collected on April 2020 and obtained from an indwelling arterial line. Patients' data were collected as part of the STORM study (Spallanzani Institute approval number 84/2020; NCT04424992). Deferred written informed consent was obtained from all participants.

As controls, we included personnel working at the European Institute of Oncology (IEO), that during the SARS-CoV-2 pandemic was involved in the prospective study SOS-COV2 (IEO 1271) (details and results of the study are published in [29]). In particular, 12 individuals scored positive to SARS-CoV-2 infection by qPCR (Swab+, Table 1) and/or by detection of circulating IgGs (IgG+, Table 1). All infected individuals reported pauci- or no-symptoms correlated to SARS-CoV-2 infection and we defined them as asymptomatic positive (asymPOS). 7 individuals tested negative to SARS-CoV-2 infection and were healthy (NEG, Table 1). Ethical approval was granted by the IEO ethical committee (IEO 1271) and written informed consent was obtained from the participants. Peripheral blood samples were collected during the first wave of the SARS-CoV-2 pandemic in Italy from June to September 2020. As controls, we took advantage of the SOS-COV2 study at our Institute in order to avoid as many confounding variables as possible, including demographic similarities and/or exposure risks. Indeed, samples from both cohorts were collected during the same phase of the SARS-CoV-2 pandemic in the same regional area (Lombardy, Italy). This minimizes also the probability of infections from different SARS-CoV-2 viral variants, that clearly showed different microbiological characteristics and, more importantly, completely different clinical impacts on the infected

**Table 1. Main clinical features of the subjects included in our study.**

| Characteristics | COV-ICU patients | | Controls | |
|---|---|---|---|---|
| | **Nr (n = 24)** | **%** | **Nr (n = 19)** | **%** |
| **AGE, Median years (Q1-Q3)** | 67 (60.25–69.75) | | 56 (37.0–61.0) | |
| 20–29 | 0 | 0 | 3 | 16 |
| 30–49 | 2 | 8.5 | 4 | 21 |
| 50–59 | 2 | 8.5 | 5 | 26 |
| 60–69 | 14 | 58 | 7 | 37 |
| 70–79 | 6 | 25 | 0 | 0 |
| **GENDER** | | | | |
| Female | 6 | 25 | 10 | 52 |
| Male | 18 | 75 | 9 | 48 |
| **Sars-CoV-2 DETECTION** | | | | |
| Swab+ IgG+ | 24 | 100 | 6 | 31.5 |
| Swab+ | 0 | 0 | 3 | 16 |
| IgG+ | 0 | 0 | 3 | 16 |
| NEG | 0 | 0 | 7 | 36.5 |
| **OUTCOME**[a] | | | | |
| Discharged alive | 16 | 67 | NA | NA |
| Deceased | 8 | 33 | NA | NA |

NA = Not applicable

[a]Applicable only to ICU patients. Controls are paucisymptomatic, asymptomatic or healthy individuals.

individuals. Moreover, we wanted to analyze controls collected before the vaccination campaign, in order to avoid possible confounding effects imposed by vaccination on the hematopoietic cell system of our individuals.

Main clinical features and demographic information of the individuals involved in our study are reported in Table 1 and S1 Table.

### Sequencing and post-processing filtering for calling of CH mutations

CH mutations were analyzed on peripheral blood mononuclear cells by error-suppressed sequencing using a custom gene panel, the CHIP-UMI Panel, which allows analyses of all protein-coding exons of the 80 genes most frequently mutated in 19 clonal hematopoiesis studies (see Supplementary Materials and methods and S1 Table in S1 File). Sequencing libraries were prepared with the SureSelect[XT HS] Target Enrichment System according to the manufacturer's instructions (Agilent Technologies). Pooled libraries were sequenced on an Illumina Novaseq 6000 with 2x100 bp paired-end reads, obtaining an average coverage of 1350X.

Analysis of the sequencing reads was performed using the Alissa software from Agilent Technologies and aligning to the GRCh38 reference human genome. We filtered for variants affecting the coding sequence: Indels and non-synonymous, stop-gain and stop/start loss SNVs. We filtered for germline polymorphisms, removing any variant reported in any population database with a frequency >0.005 by Alissa Interpret. We next applied a serial of post-processing filters in order to remove further putative germline polymorphisms and potentially false-positive variants introduced by sequencing artifacts. In particular, we used as reference a cohort of 130 samples obtained from individuals enrolled in different studies designed in IEO for the analysis of CH (see Supplementary Materials and methods in S1 File for details). We, finally, validated all variants passing these sequential filters by manual inspection on Integrative Genomics Viewer. The detailed list of mutations identified is reported in S2 Table.

### Definition of potential driver CH mutations (CH-PD)

We classified as potential drivers (PD) all CH mutations identified in our study according to the criteria described in [30–32]. In particular, for classification of somatic variants as drivers we used the criteria described in [31]. Moreover, we classified genes as oncogene or tumor suppressor genes according to OncoKB, Cancer Gene Census and other scientific literature and, then, we considered as driver: i) any truncating mutation (nonsense, essential splice site or frameshift indel) in known tumor suppressor genes; ii) any somatic mutation identified in >10 cases in the "haematopoietic and lymphoid" category in COSMIC or in the 8 studies on Myeloid neoplasms of cBioportal (see Supplementary Materials and methods in S1 File for details); iii) any somatic mutation identified in >20 cases in any other category in COSMIC or in all PanCancer Studies of cBioportal.

### Statistical analysis

Statistical analysis for comparison between disease groups was performed using GraphPad Prism software and the non-parametric two-sided Mann–Whitney U test or the one-tailed Z-test. The Mantel–Cox log-rank test was used to compare survival rates. p values <0.05 were considered significant.

### Results

To analyze the presence of CH in patients with COVD-19 disease, we performed a high-sensitivity, error-suppressed analyses of CH mutations in 24 hospitalized COVID-19 patients

(COV-ICU) and 19 controls, including 12 SARS-CoV-2-positive individuals with asymptomatic infection (asymPOS) and 7 -negative (NEG). Demographic and clinical features of the enrolled individuals are reported in Table 1 and S1 Table. The asymPOS group included the youngest individuals, with the oldest ones all in the COV-ICU cohort. Age of the negative individuals (NEG), instead, was comparable to that of the COV-ICU patients (S1 Fig in S1 File). Despite significant differences in median age, our analysis of CH mutations showed only a modest difference in numbers of mutations identified in the peripheral blood of NEG versus asymPOS individuals, and no significant differences were observed concerning the VAF and genes targeted by CH-mutations (S2 Fig in S1 File). Thus, negative and asymptomatic individuals (n = 19) were considered as a single control group and analyzed together with respect to the group of the COV-ICU patients (n = 24).

The prevalence of CH was 83.3% (20/24) and 74% (14/19) in the COV-ICU and control groups, with only 4 and 5 individuals harboring no CH mutations, respectively. Collectively, we identified 117 CH mutations in 47 genes: 71 in the COV-ICU patients and 46 in the controls, with a median VAF of 1.1% (min 0.5, max 29.5%) and 0.95% (min 0.6, max 5.9%), respectively (Table 2). The vast majority of mutations, 77.5% for COV-ICU patients and 89% for controls, had a VAF <2% (Table 2). As for the CH prevalence, neither number or VAF of CH mutations showed significant differences between COV-ICU and controls (Fig 1A and 1B, Table 2). Likewise, number of individuals harboring >1 CH mutation was similar between the two groups. Indeed, $\geq$ 3 or more mutations were identified in 12/20 (60%) COV-ICU patients and 9/14 (64%) in controls, with maximum of 9 mutations in one patient in the COV-ICU group and 6 in one control individual (Fig 1C and Table 2). In agreement with the association of CH prevalence and expansion of CH clones with age [3–5], we observed an increase in the number of CH mutations *per* patient and their VAF with age. However, these trends were again not significantly different between the two groups (Fig 2). Finally, the mutational landscapes of the two groups were also very similar (S3 Fig in S1 File). Restricting our analysis exclusively to mutations with VAF $\geq$ 2%, we scored a CH prevalence of 42% (10/24) and 21% (4/19) in COV-ICU patients and controls, respectively, in accordance with previous studies [22, 23, 26, 28]. However, this difference was not statistically significant, as for the number (17 vs 5) and the median VAF (2.8% vs 2.6%) of CH mutations in the two groups, confirming the results obtained from the analysis of our complete dataset of CH variants.

To assess their pathogenetic potential, we first annotated the identified CH mutations as putative drivers (CH-PD), according to previously described criteria [30–32] and as detailed in Materials and methods. Strikingly, we scored higher numbers and significantly higher percentages of CH-PD mutations in COV-ICU patients (22/71 mutations, 31%), as compared to controls (8/46 mutations; 17%) (Fig 3A; p = 0.044). To investigate if the identified CH mutations could potentially increase the risk of developing subsequent pathologies (hazardous mutations), we searched for their frequency in two datasets of mutations containing, respectively, all CH mutations published insofar, and the mutations reported in myeloid neoplasms in cBioPortal (https://www.cbioportal.org/, see Supplementary Materials and methods in S1 File for details). Notably, the COV-ICU cohort contained more hazardous mutations (n = 16; 21%), as compared to controls (n = 3; 6.5%) (p = 0.016; see Table 2 for details).

Finally, we analyzed clonal complexity of the CH-PD mutations in the COV-ICU and control groups. None of the control individuals harboring CH-PD mutations showed >1 mutation (0/8; Table 2). Strikingly, 8 of the 12 COV-ICU patients with CH-PD mutations (67%) showed instead >1 mutation, with two patients harboring 3 of them (Fig 3C and Table 2). Most CH-PD had a low VAF, with a median of 1.35% (min 0.7%, max 29.5%; Fig 3B and Table 2). Individual CH-PD mutations in COV-ICU patients showed a higher, albeit statistically not significant, VAF than those in controls (Fig 3B). Importantly, the higher clonal

**Table 2. CH mutations identified in our cohort.**

| | COV-ICU PATIENTS | | | | | CONTROLS | | | | |
|---|---|---|---|---|---|---|---|---|---|---|
| ID | AGE (years) | MUTATION | Driver CH-mutation | VAF (%) | Nr Mutations | ID | AGE (years) | MUTATION | Driver CH-mutation | VAF (%) | Nr Mutations |
| ICU01 | 68 | / | / | / | 0 | COV01 | 38 | / | / | / | 0 |
| ICU02 | 60 | GATA2 p.F173L | no | 1 | 1 | COV02 | 36 | / | / | / | 0 |
| ICU03 | 63 | CARD11 p.I961L | nd | 1,1 | 2 | COV03 | 38 | EP300 p.N1206K | no | 0,7 | 1 |
| | | NOTCH3 p. N247H | nd | 2,4 | | COV04 | 25 | / | / | / | 0 |
| ICU04 | 60 | PHF6 p.D264Y | no | 0,8 | 1 | COV05 | 66 | / | / | / | 0 |
| ICU05 | 69 | ATM p. H1264Lfs*5 | yes | 0,7 | 4 | COV06 | 37 | TET2 p.I1873L | yes | 5,1 | 4 |
| | | STAT3 p.L533V | no | 1,4 | | | | CARD11 p.M639L | nd | 1,9 | |
| | | ASXL1 p.A637V | no | 0,7 | | | | CUX1 p.L595R | no | 0,7 | |
| | | ASXL1 p. E635Rfs*15 | yes | 10,4 | | | | SMC1A p.I106L | no | 1,3 | |
| ICU06 | 68 | DNMT3A p. R749C | yes | 0,7 | 2 | COV07 | 26 | / | / | / | 0 |
| | | SF3B1 p.L25R | no | 0,7 | | COV08 | 29 | KMT2D p.E3007D | no | 1 | 3 |
| ICU07 | 68 | / | / | / | 0 | | | EP300 p.I338L | no | 1,1 | |
| ICU08 | 68 | PIK3CA p. Q1033H | no | 1,3 | 4 | | | SMC1A p.T8P | no | 1,1 | |
| | | ATM p.H1802Q | no | 1,1 | | COV09 | 51 | DNMT3A p. P264Gfs*15 | yes | 0,6 | 2 |
| | | KMT2D p.Q4347P | no | 1,6 | | | | NOTCH3 p.D331E | nd | 0,7 | |
| | | STAT3 p.Y22D | no | 0,9 | | COV10 | 69 | RICTOR p.I111L | nd | 1,1 | 5 |
| ICU09 | 62 | TET2 p.Q1435P | yes | 0,9 | 2 | | | STAT3 p.D661Y | yes | 0,9 | |
| | | BRCC3 p.W130* | yes | 1,3 | | | | STAT3 p.Q274P | no | 0,7 | |
| ICU10 | 58 | NF1 p.K598Q | no | 3,1 | 4 | | | GNAS p.G2R | no | 1,1 | |
| | | CUX1 p.L467* | yes | 2,4 | | | | BCORL1 p.R1540L | no | 0,8 | |
| | | BRAF p.K618Q | yes | 0,9 | | COV11 | 61 | PIK3CA p.I713L | nd | 1,5 | 2 |
| | | ASXL1 p.Q141P | no | 0,9 | | | | ATM p.P1235H | no | 1,2 | |
| ICU11 | 34 | CBL p.L295R | no | 2,5 | 1 | COV12 | 61 | SF3B1 p.M613R | yes | 1,1 | 6 |
| ICU12 | 70 | DNMT3A p. W753R | yes | 0,8 | 3 | | | TET2 p.H672P | no | 0,8 | |
| | | TET2 p.R1465* | yes | 1,8 | | | | KMT2D p.Q4347P | no | 1,7 | |
| | | CBL p.Y774* | no | 1,9 | | | | NOTCH3 p. D1936E | nd | 0,9 | |
| ICU13 | 71 | GNAS p.P438A | no | 2,2 | 5 | | | SMC1A p.M1003I | no | 0,8 | |
| | | GNAS p.S455A | no | 3,7 | | | | STAG2 p.P1143T | no | 0,8 | |
| | | ATRX p.V2132L | no | 0,9 | | COV13 | 59 | CUX1 p. N1204Tfs*6 | yes | 0,6 | 2 |
| | | PHF6 p.M1? | no | 1,5 | | | | KMT2D p.K2032Q | no | 0,9 | |
| | | GNB1 p.K57E | yes | 0,7 | | COV14 | 56 | CBLB p.K685N | nd | 0,8 | 1 |

(*Continued*)

**Table 2.** (Continued)

| ID | AGE (years) | MUTATION | Driver CH-mutation | VAF (%) | Nr Mutations | ID | AGE (years) | MUTATION | Driver CH-mutation | VAF (%) | Nr Mutations |
|---|---|---|---|---|---|---|---|---|---|---|---|
| **COV-ICU PATIENTS** | | | | | | **CONTROLS** | | | | | |
| ICU14 | 74 | NOTCH2 p.C842W | nd | 1,5 | 9 | COV15 | 60 | GNB1 p.L192R | no | 1,2 | 4 |
| | | TET2 p.I750Rfs*62 | yes | 3,3 | | | | DNMT3A p.E561* | yes | 0,9 | |
| | | TET2 p.N1266T | yes | 29,5 | | | | CARD11 p.T43P | nd | 2,2 | |
| | | BIRC3 p.R448L | nd | 0,9 | | | | CREBBP p.S18R | no | 0,8 | |
| | | KMT2D p.P619Q | no | 0,7 | | COV16 | 65 | KMT2D p.H3337P | no | 1,8 | 3 |
| | | FLT1 p.Y268H | nd | 3 | | | | BCOR p.M1259I | no | 1,9 | |
| | | SETBP1 p.L1491R | nd | 2,8 | | | | BCOR p.K1195N | no | 0,9 | |
| | | ATRX p.P672L | no | 1 | | COV17 | 58 | DNMT3A p.P777S | yes | 1,3 | 5 |
| | | TET2 p.E432Tfs*9 | yes | 0,6 | | | | SETD2 p.N923K | no | 0,7 | |
| ICU15 | 46 | / | / | / | 0 | | | NOTCH1 p.I807M | no | 2 | |
| ICU16 | 66 | DNMT3A p.I705Mfs*74 | yes | 2,2 | 2 | | | AXL p.P879L | nd | 5,9 | |
| | | PPM1D p.S468* | yes | 9,7 | | | | KDM6A p.P417R | no | 0,8 | |
| ICU17 | 72 | SMAD4 p.N306K | no | 2,5 | 1 | COV18 | 56 | SETDB1 p.L986W | nd | 0,9 | 4 |
| ICU18 | 62 | NOTCH2 p.Q677P | nd | 0,7 | 8 | | | SETD2 p.E1156D | no | 0,8 | |
| | | ASXL2 p.V579G | no | 1,6 | | | | SMC1A p.L1116R | no | 1,1 | |
| | | TET2 p.H1727Q | no | 0,7 | | | | ASXL2 p.T1302P | no | 1,2 | |
| | | FBXW7 p.L10R | no | 1,2 | | COV19 | 61 | DNMT3A p.E442Gfs*4 | yes | 2,6 | 4 |
| | | CALR p.E255D | no | 0,7 | | | | RICTOR p.N848S | nd | 1,2 | |
| | | ZRSR2 p.C326F | no | 0,8 | | | | KRAS p.S89* | no | 0,7 | |
| | | BCOR p.Y972D | no | 1,6 | | | | ATRX p.M1920R | no | 0,9 | |
| | | STAG2 p.A969D | no | 0,9 | | | | | | | |
| ICU19 | 61 | / | / | / | 0 | | | | | | |
| ICU20 | 59 | TP53 p.M237I | yes | 0,7 | 6 | | | | | | |
| | | FBXW7 p.L10R | no | 1,5 | | | | | | | |
| | | SETBP1 p.I923L | nd | 1 | | | | | | | |
| | | SMAD4 p.A319D | no | 0,7 | | | | | | | |
| | | JAK3 p.L19V | no | 0,8 | | | | | | | |
| | | NOTCH2 p.C19W | yes | 2,3 | | | | | | | |
| ICU21 | 65 | DNMT3A p.R899C | yes | 1,8 | 3 | | | | | | |
| | | SMC1A p.R273P | no | 0,8 | | | | | | | |
| | | BRCC3 p.Q299* | yes | 1 | | | | | | | |
| ICU22 | 70 | NOTCH2 p.E1223D | nd | 1,9 | 4 | | | | | | |
| | | SETD2 p.L64R | no | 0,7 | | | | | | | |
| | | ASXL1 p.G587Rfs*32 | yes | 0,5 | | | | | | | |
| | | NOTCH1 p.C1209G | no | 0,7 | | | | | | | |
| ICU23 | 74 | JAK2 p.H172P | no | 1 | 3 | | | | | | |
| | | EP300 p.Q2048P | no | 1,9 | | | | | | | |
| | | STAG2 p.D381Y | no | 0,9 | | | | | | | |

(Continued)

**Table 2.** (Continued)

| | COV-ICU PATIENTS | | | | | CONTROLS | | | | | |
|---|---|---|---|---|---|---|---|---|---|---|---|
| **ID** | **AGE (years)** | **MUTATION** | **Driver CH-mutation** | **VAF (%)** | **Nr Mutations** | **ID** | **AGE (years)** | **MUTATION** | **Driver CH-mutation** | **VAF (%)** | **Nr Mutations** |
| ICU24 | 68 | DNMT3A p. R326H | yes | 0,7 | 6 | | | | | | |
| | | SF3B1 p.R549C | yes | 2,4 | | | | | | | |
| | | SF3B1 p.L25R | no | 1,4 | | | | | | | |
| | | KMT2D p.G4844S | no | 1,7 | | | | | | | |
| | | ZRSR2 p.L37R | no | 1,5 | | | | | | | |
| | | ASXL1 p. G646Wfs*12 | yes | 1,4 | | | | | | | |
| VAF: Variant allelle frequency | | | | | | | | | | | |
| Common to CHIP-mutations | | | | | | | | | | | |
| Common to mutations in Myeloid Neoplasms according to cBioportal | | | | | | | | | | | |
| Common to both CHIP-mutations and Myeloid Neoplasms according to cBioportal | | | | | | | | | | | |
| nd: not defined yet | | | | | | | | | | | |

complexity, higher numbers and VAF of CH-PD mutations in the COV-ICU group vs. controls appeared to be age-independent. In the age group 50–59 years old, we observed 2/2 (100%) patients with a total of 3 driver mutations with a mean VAF of 1.8% (range 0.7–2.4%) in the COV-ICU group, as compared to 3/5 (60%) individuals with 3 driver mutations with mean VAF of 0.8% (range 0.6–1.3%) in the control group. As well, in the age group 60–69 years old, we observed 6/11 (55%) patients with 12 driver mutations and mean VAF of 2.8% (range 0.7–10.4%) in the COV-ICU and 4/6 (67%) with 4 driver mutations and average VAF of 1.4% (range 0.9–2.6%) in the controls (Fig 3C and Table 2). Nonetheless, the presence of driver mutations in the COV-ICU patients did not seem to significantly affect their survival or their clinical parameters (S1 Table and S4 Fig in S1 File).

## Discussion

To our knowledge, this is the first study that analyzes the impact of CH on COVID-19 disease with a limit of sensitivity equal to 1% of nucleated blood cells, which is considerably lower to the 4% currently used to define CHIP operatively. Consistently, as compared to the published studies [22–24, 26–28], we identified significantly higher numbers of CH mutations *per* patient (overall, 2.8 vs <1 mutations) and higher prevalence of CH (overall, >80% vs <50%) and found CH clones also in individuals with <40 years. These data are in agreement with previous studies of CH at high sensitivity in non-COVID individuals, which also reported a higher prevalence of CH in the population, as compared to standard NGS approaches, especially at younger ages [30, 33–36].

Nonetheless, our data showed similar prevalence of CH, size of CH clones and mutational landscape in COVID-19 patients and controls, ruling out a positive association between CH and incidence and/or aggravation of the infection from SARS-CoV-2, in line with results reported in Hameister et al. [26], Petzer et al. [27] and Zhou et al. [28]. A recent study showed an association between presence of myeloid CH clones and increased risk of mortality due to COVID-19. This association, however, was significant only in the age group of 75–84 years old individuals [37]. In our cohort, there are no COVID-19 patients ≥75 years old. Therefore, we cannot exclude that the absence of association between CH and outcome of the COVID-19

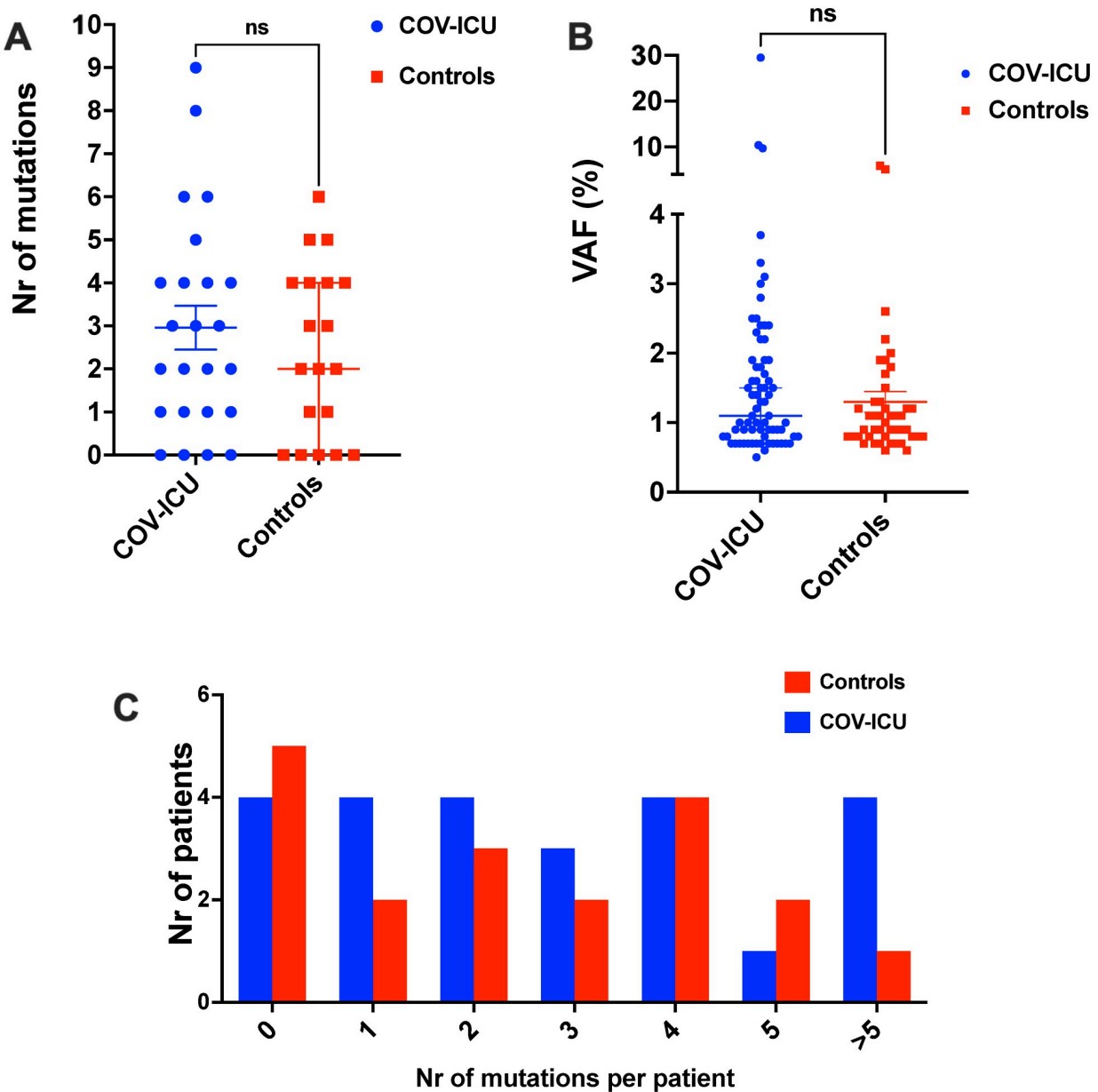

**Fig 1. Clonal hematopoiesis in COVID-19 patients.** A) Number of CH mutations identified in each individual. B) Variant allele frequency (VAF) of the CH mutations identified. C) Number of patients with the indicated number of CH mutations in our cohort. The horizontal line indicates the mean and the whiskers the SEM. Statistical significance was calculated using Mann–Whitney U test. COV-ICU, patients with severe COVID-19; Controls, control group; ns, not-significant.

disease in our study is due to the younger age of our patients. Nonetheless, interestingly, Del Pozo-Valero et al. reported that CH variants classified as pathogenic or likely pathogenic were significantly more represented in the COVID-19 patients with increased risk of mortality [37]. This is in line with our observation, that COVID-19 patients with severe disease harbor more CH-PD and hazardous mutations, as discussed below.

More than 80% of the mutations we identified had VAF<2%. The biological meaning and clinical relevance of these small clones remain to be elucidated. They could simply represent a

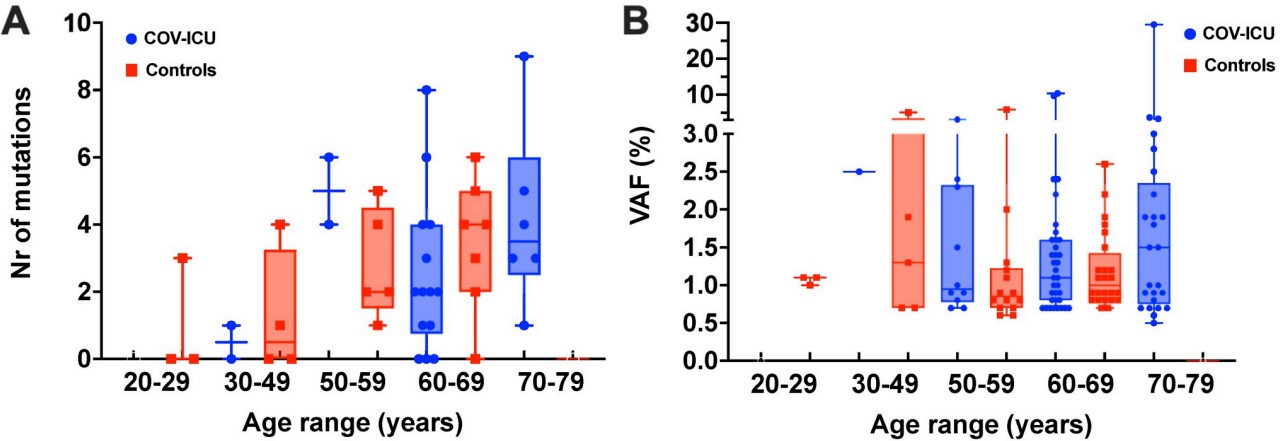

**Fig 2. Distribution of CH mutations by age in our cohort.** A) Number of CH mutations per individual according to their age. B) Variant allele frequency (VAF) of CH mutations in individuals according to their age. COV-ICU, patients with severe COVID-19; Controls, control group. Boxes define the 25th and the 75th percentiles; horizontal line within the boxes indicates the median and whiskers define the Min to Max002E.

transient modification of the genomic landscape of the peripheral blood of healthy individuals, highly dynamic and constantly changing. Mutated clones likely change in time quite frequently and, without a specific environmental pressure, most of these mutated clones at low frequency could simply be envisioned as passengers with no particular consequences for the fitness of the cells harboring them, as suggested by recent reports [38–40]. Regardless of mechanisms and duration of the fitness advantage conferred by low-VAF mutations, their presence might impact different physiological functions, including immunity, thus influencing the penetrance of different disease phenotypes.

It has been reported that CH clones with different mutations expand with different growth rates, which are dependent both on the mutated gene and the specific aminoacidic change: i.e. mutations in DNMT3A show a slower clonal expansion (5% per year) compared to mutations in TET2 (10% per yr) or splicing factors (15–20% per yr). Moreover, mutation fitness appears not to be constant over the life time of an individual: some clones grow more rapidly early in life and then their growth rate decreases during old age (clones with mutations in DNMT3A, BRCC3, TP53), while other clones show no deceleration (clones with mutations in U2AF1, SRSF2$^{P95H}$, IDH1) and TET2 mutations show a quite constant growth rate and overtake clonal hematopoiesis later in life [40]. Considering that the majority of the clones we identified have a VAF below the clinical threshold of 2% and we found the majority of CH-PD mutations in DNMT3A (no in the hotspot R882 position, which is the DNMT3A mutated position that confers the highest fitness to clonal growth), TET2 and ASXL1, we expect that clonal expansion in our patients would require a time frame of several years. The design and development of specific trials with long-term longitudinal follow-up would be instrumental in order to score clonal expansion and define if the CH-PD mutations we identified are *bona fide* markers of disease progression.

In our cohort, we found 15 CH-PD mutations (14 in COV-ICU and 1 in controls) in common with mutations identified in myeloid neoplasms in the cBioPortal database. Interestingly, some of these mutations had low VAF in patients with myeloid neoplasms, similar to the VAF scored in our study. Namely, DNMT3A p.R899C and BRCC3 p.Q299* have been reported with a VAF = 2% in patients with MDS and were found with a VAF equal to 1.8% and 1% in our COV-ICU patients, respectively. The hotspot mutations GNB1 p.K57E has been reported

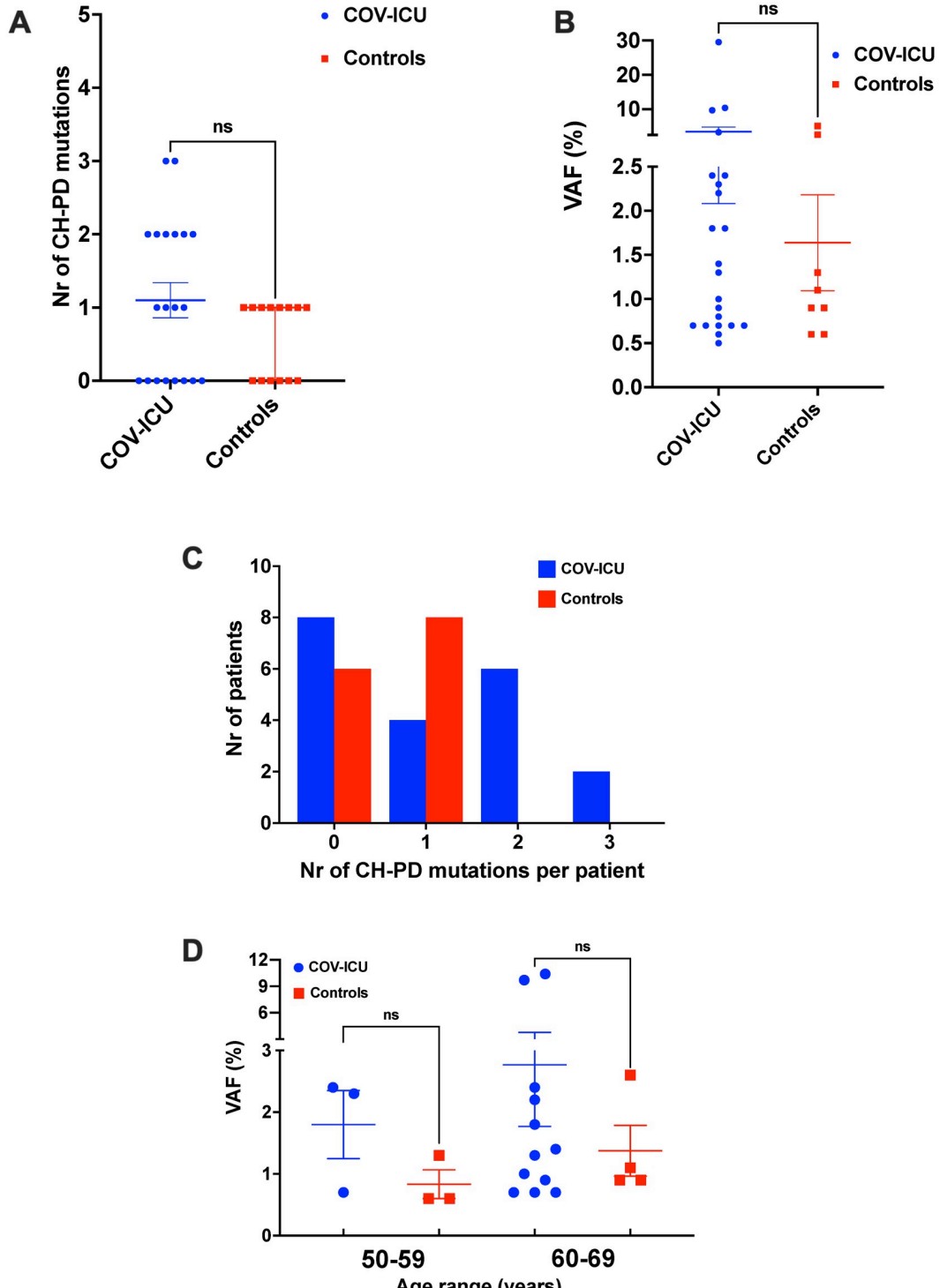

**Fig 3. Distribution of potential driver CH mutations in our cohort.** A) Number of CH-PD mutations identified in each individual. B) Variant allele frequency (VAF) of the CH-PD mutations identified. C) Number of patients with the indicated number of CH-PD mutations in our cohort. D) VAF of CH-PD mutations in individuals according to their age. The horizontal line indicates the mean and the whiskers the SEM. Statistical significance was calculated using Mann–Whitney U test. COV-ICU, patients with severe COVID-19; Controls, control group; CH-PD, potential driver mutation in CH.

in several patients with MDS with VAF = 2%. PPM1D truncating variant p.S468*, found with a VAF = 9.7% in ICU16, has been reported with a VAF = 3 and 4% in two patients with MDS. TET2 p.I750Rfs*62 have been reported in a patient with CMML. Furthermore, in a recent longitudinal study, it has been shown that fast growing clones with potentially harmful variants were originally identified has clones with a VAF<2% [39]. Altogether these data suggest that clinically relevant variants can be found at VAF that do not reach the clinical threshold of CHIP and that we need to favor more inclusive analysis of CH because significant information can be revealed by low frequency mutations.

Mutations in different CH-genes have been associated to distinct risks of development of subsequent pathologies, in particular acute myeloid leukemia (AML) and coronary heart disease (CHD). For example, mutations in *TP53* or in splicing factors (*U2AF1*, *SF3B1*, *SRSF2*) are linked to a particularly high risk of developing AML. Considering CHD, instead, mutations in *DNMT3A*, *TET2* and *ASXL1* double the relative risk, while mutations in *JAK2* increase the relative risk up to 12 folds (reviewed in [6]). Thus, it appears that the type of CH mutations impact on the evolution of potential downstream diseases. In our cohort, in patients with severe COVID-19 disease we observed a significantly higher percentage of CH-PD mutations, most of which (16/22, 73%) in common with datasets of hazardous mutations. Moreover, in COV-ICU patients the clonal complexity for these mutations was higher than in controls, suggesting that their presence might have a clinical impact. We cannot formally exclude the possibility that these CH-PD mutations could confer an increased risk of faster degenerating COVID-19 disease or prolonged illness, may be affecting the host immune responses and their antiviral defenses. However, most notably, in the COV-ICU patients we did not score an impact of the presence of these potential driver mutations on patient survival or clinical parameters associated with inflammation.

Even if the numbers of our cohort are too small to draw definitive conclusions, the lack of an impact of high-frequency CH-PD on clinical-biological parameters argues against an association between CH and COVID-19 prognosis, suggesting that the COVID-19 disease might instead select for hazardous CH-PD mutations. Systemic factors associated to severe disease, such as increased production of inflammatory cytokines, could favor the positive selection of clones with a selective advantage. Under this scenario, clonal expansion could be reversible and the CH clones disappear after clinical recovery from COVID-19, or, alternatively, once a mutation conferring proliferative advantage is acquired, the clones may continue to expand independently from the resolution of the COVID-19 disease, thus increasing the risk of hematological disease or other aging-associated morbidities. To our knowledge, up to date, only three studies analyzed the dynamic of CH clones in paired samples from the same individual at different follow-up: in 9 COVID-19 patients from 7 to 16 days apart, in 8 patients during a 6 months follow-up, and in 54 critically ill patients tested 8 days apart [23, 27, 28]. None of the studies reported major changes in VAF of the detected CH variants at the different time points analyzed, suggesting no mayor effects on clonal expansion [23, 27, 28].

In conclusion, our data seem to suggest that the COVID-19 disease could influence the clonal composition of the peripheral blood of COVID-19 patients with severe disease. Nonetheless, our study uses a relatively small sample size. It will indispensable to perform appropriate further studies on larger patient cohorts in order to be able to validate and generalize our conclusions. Moreover, we analyzed the PB clonal composition at a single time point. It will be necessary to consider longitudinal approaches to track VAF variations over time with long periods of follow-up in order to assess if the COVID-19 disease could have a long-term impact on the evolution of CH in people that experienced severe COVID-19 and long-term consequences on patient outcomes.

## Supporting information

**S1 File.**
(DOCX)

**S1 Table. Clinical parameters of COVID19 patients admitted to ICU.**
(XLSX)

**S2 Table. Detailed list of CH-mutations identified in our study.**
(XLSX)

## Acknowledgments

We thank IEO Covid Team and all IEO personnel that participated to sample collections and laboratory measurements for the SOS-COV2 study in IEO and the Genomics Unit of the Department of Experimental Oncology of IEO for sequencing of the CH libraries.

## Author Contributions

**Conceptualization:** Myriam Alcalay, Roberto Orecchia, Gioacchino Natoli, Pier Giuseppe Pelicci.

**Data curation:** Chiara Ronchini, Chiara Caprioli, Francesco Furio D'Amico.

**Formal analysis:** Chiara Ronchini, Chiara Caprioli, Francesco Furio D'Amico.

**Funding acquisition:** Myriam Alcalay.

**Resources:** Gianleo Tunzi, Emanuela Colombo, Marco Giani, Giuseppe Foti, Donatella Conconi, Marialuisa Lavitrano, Rita Passerini, Luca Pase, Silvio Capizzi, Fabrizio Mastrilli.

**Supervision:** Rita Passerini, Luca Pase, Silvio Capizzi, Fabrizio Mastrilli, Roberto Orecchia, Gioacchino Natoli, Pier Giuseppe Pelicci.

**Visualization:** Chiara Ronchini.

**Writing – original draft:** Chiara Ronchini.

**Writing – review & editing:** Chiara Ronchini, Chiara Caprioli, Myriam Alcalay, Gioacchino Natoli, Pier Giuseppe Pelicci.

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
