## [Decision Letter · Decision Letter 0]

21 Jun 2023

PONE-D-23-04485High-sensitivity Analysis of Clonal Hematopoiesis Reveals Increased Clonal Complexity of Potential-Driver Mutations in Severe COVID-19 PatientsPLOS ONE

Dear Dr. Ronchini,

Thank you for submitting your manuscript to PLOS ONE. After careful consideration, we feel that it has merit but does not fully meet PLOS ONE’s publication criteria as it currently stands. Therefore, we invite you to submit a revised version of the manuscript that addresses the points raised during the review process.

We look forward to receiving your revised manuscript.

Kind regards,

Antonio Solimando

Academic Editor

PLOS ONE

Journal Requirements:

Additional Editor Comments:

The authors are asked to revise the manuscript according to the reviewer's comments.

Reviewers' comments:

Reviewer's Responses to Questions

**Comments to the Author**

1. Is the manuscript technically sound, and do the data support the conclusions?

Reviewer #1: Yes

Reviewer #2: Yes

2. Has the statistical analysis been performed appropriately and rigorously? 

Reviewer #1: Yes

Reviewer #2: Yes

3. Have the authors made all data underlying the findings in their manuscript fully available?

Reviewer #1: Yes

Reviewer #2: Yes

4. Is the manuscript presented in an intelligible fashion and written in standard English?

Reviewer #1: Yes

Reviewer #2: Yes

5. Review Comments to the Author

Reviewer #1: Doctors Chiara Ronchini, Pier Giuseppe Pelicci, and colleagues present a study of the prevalence of clonal hematopoiesis (CH)(down to a reported variant allele detection threshold of 0.5%) in 24 patients admitted to the ICU for COVID-19 (COV-ICU), and in 19 controls that include healthy subjects and those with asymptomatic, documented SARS-CoV2 infection. Major findings included the high prevalence of CH in both groups, mainly with VAF <2%, as expected by the error-corrected sequencing approach and the few previous studies that have found CH to be common in adults at VAF less than the CHIP threshold of 2%. When considering CH variants more strongly associated with driving clonal expansion and/or myeloid malignancy (CH-PD or "hazardous mutations"), the authors found a significantly higher prevalence in COV-ICU patients, as compared to controls, along with increased features of clonal complexity. Finally, the authors found no significant impact of the presence of CH-PD on clinical outcomes of ICU patients, including the probability of survival, or blood counts, clinical indicators of hemostasis or oxygen measures.

There have been several publications relating to the prevalence and clinical impact of CH in COVID-19 and the authors acknowledge these. The novelty lies mostly in the application of error-suppressed sequencing and the analysis of CH clones at lower mean VAF than previous published studies.

The major limitation, which the authors also acknowledge, is that the number of participants in their cohort is small and that this limits the ability to draw definitive conclusions. Nevertheless, their observations call for larger studies to examine (or re-examine) the potential impact of more ubiquitous, lower-VAF level CH, and prospective analysis of longer-term outcomes of COVID patients who survive ICU admission, with and without detectable CH.

These additional comments are meant to improve the quality of the manuscript:

1. It seems that the custom, error-corrected sequencing approach ("CHIP-UMI") is new and its application has not been previously peer-reviewed or published before. Otherwise, please clarify. If so, careful consideration should be given to method validation. From experience, "error-correction" is a misnomer and "error-suppression" is probably a more suitable description for such methods. The authors do present in the Supplemental Materials and Methods a limited and descriptive summary of the "validated...performance" of the custom panel. Why not show some of this data in the supplement? My concern is mainly with artifacts and "Likely-FalsePositive" findings that remain even after error suppression and filtering. Their list of CH variants, while appropriately labelled with a dedicated "Driver CH mutations" column, and colour codes to reflect levels of evidence, may still contain artifacts/errors. For example, in this small sample set the following variants appear twice: ASXL2 c.3900C>G, FBXW7 c.29T>G, KMT2D c.13040A>C, SF3B1 c74T>G. For the most part, these do not appear to be common CH or cancer-associated variants. Also, the ATRX c.5424T>G variant suspiciously appears 3 times, and while listed as "yes" for "Driver CH mutation", this reviewer cannot find reference to this ATRX variant in the CH literature. Finally, as the authors may be aware, the ASXL1 c.1934dupG (p.G646Wfs*12) variant can be contentious, and has been reported as both an artifact and true somatic variant, with the likelihood of the latter increasing with higher VAF (see PubMedID: PMID: 36652671). In this manuscript, the reported VAF is only 1.4%. Can the authors please review these variants and provide justification for their inclusion, perhaps also including an independent means (such as ddPCR) to confirm their validity, or remove them or provide further qualifications about limitations of this method?

2. What is the prevalence of CHIP (as defined by VAF >= 2%) in the cohorts? Without performing the analysis myself, it is not clear. The authors do state that "the vast majority of mutations...had a VAF <2%" but what about considering on a per-patient basis? I'm assuming the numbers will be too small for meaningful analysis of outcomes, but more information about traditional CHIP will allow a better comparison with previously published studies. Related to this, as compared to other studies (including Bolton et al.), the authors find here a higher proportion of TET2 variants (where Bolton and others have found more dominance of DNMT3A). Could this relate to the patient demographics, such as the number of patients with cardiovascular co-morbidity)? This may be worth discussing.

3. The lack of comparator for the 70s age group might be worth acknowledging. A recent study by Del Pozo Valero et al. found that myeloid-CH significantly increased risk of COVID-19 mortality among individuals ≥75 years, but not among those in their 60s (please see PMID: 36184726). Perhaps the lack of significance found here could be partly attributable to the relatively younger age groups that were compared. Please consider citing and discussing this study and potential limitation.

4. The lack of impact of CH on clinical outcomes and parameters otherwise aligns with other studies, and the authors mention this in the manuscript. However, there may be other considerations regarding the statistically significant association of PD-CH in severe COVID-19. The authors mention "COVID-19 disease might instead select for hazardous CH-PD mutations". It might be pertinent to mention and consider the converse - that these types of PD-CH mutations could confer greater risk of severe disease, even if they’re not directly impacting prognosis while in the ICU (i.e., greater implications on host antiviral defenses rather than prolonged illness).

5. Perhaps consider mentioning earlier in the METHODS, Patient cohort section, that patient demographic information (such as age, sex etc.) is available and point the reader to the appropriate tables.

6. Regarding patient demographics, the supplemental table includes the exact date of birth and date of admission. This may be too identifying. The age (in years) at presentation may be sufficient, and the authors already state that patients presented/samples were collected in April 2020. Please consider if this type of (potentially identifying) information should be removed from the supplement.

7. Gene names should be in italics (as an example, in the Introduction, line 59; but please check elsewhere).

Reviewer #2: The authors re-evaluate the link between clonal hematopoiesis and COVID-19 severity by increasing the detection sensitivity of the defining somatic mutations (threshold 0.5%) in 24 patients with severe disease, 12 asymptomatic patients and 9 non-infected donors. The tested gene panel includes 80 genes, the coverage is 1350X, COSMIC and cBioportal are used to identify driver genes with a VAF that, in more than 80% of cases, was in the 0.5-2% range. The total number of detected mutations was similar in both group. Then, potential driver mutations and clonal complexity were analyzed, and these two parameters were higher in severe patients, without significant impact on clinical and biological parameters and disease outcome in this small cohort. Altogether, this study further argue for a limited impact of clonal hematopoiesis on severe COVID-19 outcome. The manuscript is clearly written and complements a series of previous analyses with various conclusions in small cohorts. A meta analysis will be needed for solid conclusions regarding the outcome. This report is a useful contribution to prepare this analysis.

In the introduction, the authors may stick to WHO 2022 classification of CHIP (Khoury et al, Leukemia 2022): CHIP refers to somatic mutations of myeloid malignancy-associated genes detected in the blood or bone marrow at a VAF ≥ 2% (≥4% for X-linked gene mutations in males) in individuals without a diagnosed hematologic disorder and without unexplained cytopenia.

The WHO classification also defines clonal cytopenia of undetermined significance (CCUS) as CHIP in the presence of one or more persistent cytopenias that are otherwise unexplained by hematologic or non-hematologic conditions and that do not meet diagnostic criteria for defined myeloid neoplasms. Did some of the patients included in this trial demonstrate a cytopenia before inclusion?

The introduction also focuses on the cardiovascular risk of CHIP but many other risks have now been identified, the latest being liver diseases (see B Ebert’s paper in Nature recently)

Since they were included in 2020, do the authors have some information regarding the long-term outcome of the survivors?

6. PLOS authors have the option to publish the peer review history of their article (what does this mean?). If published, this will include your full peer review and any attached files.

Reviewer #1: No

Reviewer #2: No

---

## [Author Response · Author response to Decision Letter 0]

4 Aug 2023

We thank the Reviewers for their valued comments and for highlighting some issues, we acknowledge that they helped us improving the quality of our manuscript.

In the following pages are our point-by-point responses (line # according to the revised manuscript without track changes). 

Reviewer #1: Doctors Chiara Ronchini, Pier Giuseppe Pelicci, and colleagues present a study of the prevalence of clonal hematopoiesis (CH)(down to a reported variant allele detection threshold of 0.5%) in 24 patients admitted to the ICU for COVID-19 (COV-ICU), and in 19 controls that include healthy subjects and those with asymptomatic, documented SARS-CoV2 infection. Major findings included the high prevalence of CH in both groups, mainly with VAF <2%, as expected by the error-corrected sequencing approach and the few previous studies that have found CH to be common in adults at VAF less than the CHIP threshold of 2%. When considering CH variants more strongly associated with driving clonal expansion and/or myeloid malignancy (CH-PD or "hazardous mutations"), the authors found a significantly higher prevalence in COV-ICU patients, as compared to controls, along with increased features of clonal complexity. Finally, the authors found no significant impact of the presence of CH-PD on clinical outcomes of ICU patients, including the probability of survival, or blood counts, clinical indicators of hemostasis or oxygen measures.

There have been several publications relating to the prevalence and clinical impact of CH in COVID-19 and the authors acknowledge these. The novelty lies mostly in the application of error-suppressed sequencing and the analysis of CH clones at lower mean VAF than previous published studies.

The major limitation, which the authors also acknowledge, is that the number of participants in their cohort is small and that this limits the ability to draw definitive conclusions. Nevertheless, their observations call for larger studies to examine (or re-examine) the potential impact of more ubiquitous, lower-VAF level CH, and prospective analysis of longer-term outcomes of COVID patients who survive ICU admission, with and without detectable CH.

These additional comments are meant to improve the quality of the manuscript:

1. It seems that the custom, error-corrected sequencing approach ("CHIP-UMI") is new and its application has not been previously peer-reviewed or published before. Otherwise, please clarify. If so, careful consideration should be given to method validation. From experience, "error-correction" is a misnomer and "error-suppression" is probably a more suitable description for such methods. The authors do present in the Supplemental Materials and Methods a limited and descriptive summary of the "validated...performance" of the custom panel. Why not show some of this data in the supplement? My concern is mainly with artifacts and "Likely-FalsePositive" findings that remain even after error suppression and filtering. Their list of CH variants, while appropriately labelled with a dedicated "Driver CH mutations" column, and colour codes to reflect levels of evidence, may still contain artifacts/errors. For example, in this small sample set the following variants appear twice: ASXL2 c.3900C>G, FBXW7 c.29T>G, KMT2D c.13040A>C, SF3B1 c74T>G. For the most part, these do not appear to be common CH or cancer-associated variants. Also, the ATRX c.5424T>G variant suspiciously appears 3 times, and while listed as "yes" for "Driver CH mutation", this reviewer cannot find reference to this ATRX variant in the CH literature. Finally, as the authors may be aware, the ASXL1 c.1934dupG (p.G646Wfs*12) variant can be contentious, and has been reported as both an artifact and true somatic variant, with the likelihood of the latter increasing with higher VAF (see PubMedID: PMID: 36652671). In this manuscript, the reported VAF is only 1.4%. Can the authors please review these variants and provide justification for their inclusion, perhaps also including an independent means (such as ddPCR) to confirm their validity, or remove them or provide further qualifications about limitations of this method?

Response: The sequencing approach we used for this study with the CHIP-UMI panel has not been published yet. The manuscript about the validation of our approach is in preparation. We added some details on the performance of the method in the Supplementary Materials and Methods section and we changed the nomenclature to error-suppression, as suggested. We are well aware of the issue of sequencing artifacts and that is why we created a database of "Likely-FalsePositive", which we are constantly updating with new sequencing data from new individuals with the CHIP-UMI panel. Since the submission of the paper to PLOSone we increased the number of sequenced samples with the CHIP-UMI from 91 to 130. 

We carefully reevaluated all CH variants identified in our study and, in particular, the ones highlighted by the Reviewer. We removed from our dataset ASXL2 c.3900C>G and ATRX c.5424 T>G, because with the new analysis they were found in ≥3 individuals and are, therefore, potential artifacts. We revisited all our data and analysis and updated all tables and figures, accordingly.

Concerning the ASXL1 c.1934dupG (p.G646Wfs*12) variant, we are aware of the conflicting interpretation as artifact or real somatic variant. However, based on our approach and what is reported in the literature, we believe that in our cohort it is a true somatic variant. From a technical point of view, for library preparation, we used the Agilent SureSelect kit. Enrichment of the region of interest is based on probe capture by hybridization and the kit provides the proof-reading DNA polymerase Herculase II Fusion DNA Polymerase. NGS protocols for libraries preparation with these characteristics have been shown to overcome the issue of detection of this variant as an artifact (PMID: 30222780). Moreover, we applied an error suppression protocol, obtaining for this position very good sequencing parameters: a coverage of 957 reads with >10 paired-reads supporting the alternative allele. From a biological point of view, the ASXL1 p.G646Wfs*12 true somatic variant were usually age-related (PMID: 36652671) and in the 94% of cases were reported in combination with other mutations with a similar VAF (median 4 somatic mutations per case, range 1±8) (PMID: 30222780). Within our entire cohort of 130 individuals, we detected this mutation exclusively in this COVID-19 patient (ICU24), who is 68 years old and harbors together with this ASXL1 variant, other 6 variants with similar VAF. In particular, two of these variants are drivers and have been reported both in CHIP and myeloid malignancies: DMT3A p.R326H and SF3B1 p.R549C. 

All other variants highlighted by the Reviewer, within our entire cohort of 130 individuals, are still found in only two patients and, interestingly, are identified only within the COVID-19 cohort, suggesting they could be specific variants for this clinical setting. All variants are well covered (mean coverage of 830 reads and mean of 20.5 reads supporting the alternative allele). According to our analytic workflow and variant filtering, we can unbiasedly remove these variants from our dataset. Finally, none of these variants is classified as driver or hazardous mutation, therefore, they do not impact the conclusions or the implications of our study. 

2. What is the prevalence of CHIP (as defined by VAF >= 2%) in the cohorts? Without performing the analysis myself, it is not clear. The authors do state that "the vast majority of mutations...had a VAF <2%" but what about considering on a per-patient basis? I'm assuming the numbers will be too small for meaningful analysis of outcomes, but more information about traditional CHIP will allow a better comparison with previously published studies. Related to this, as compared to other studies (including Bolton et al.), the authors find here a higher proportion of TET2 variants (where Bolton and others have found more dominance of DNMT3A). Could this relate to the patient demographics, such as the number of patients with cardiovascular co-morbidity)? This may be worth discussing.

Response: We added the results restricting our analysis to CH variants with VAF≥2% in the Results section (line 171 line 176). Similar to what we observed considering all CH variants, we did not score statistically significant differences between COV-ICU patients and controls. These data show that restricting our analysis to CHIP mutations has no impact on the main results and conclusions of our study. 

Concerning the number of mutations identified in DNMT3A or TET2, in the general cohort, we scored more mutations in DNMT3A compared to TET2, as reported in most studies. If the Reviewer is referring exclusively to the COVID-19 cohort, the difference we score is of 6 mutations in TET2 vs 5 mutations in DNMT3A. We think our numbers and the differences we scored are definitively too small to be able to observe significant correlations with any clinical or demographic parameters. 

3. The lack of comparator for the 70s age group might be worth acknowledging. A recent study by Del Pozo Valero et al. found that myeloid-CH significantly increased risk of COVID-19 mortality among individuals ≥75 years, but not among those in their 60s (please see PMID: 36184726). Perhaps the lack of significance found here could be partly attributable to the relatively younger age groups that were compared. Please consider citing and discussing this study and potential limitation.

Response: We discuss the results of this recent paper in the DISCUSSION section, line 216-225. 

4. The lack of impact of CH on clinical outcomes and parameters otherwise aligns with other studies, and the authors mention this in the manuscript. However, there may be other considerations regarding the statistically significant association of PD-CH in severe COVID-19. The authors mention "COVID-19 disease might instead select for hazardous CH-PD mutations". It might be pertinent to mention and consider the converse - that these types of PD-CH mutations could confer greater risk of severe disease, even if they’re not directly impacting prognosis while in the ICU (i.e., greater implications on host antiviral defenses rather than prolonged illness).

Response: We added a comment in the DISCUSSION section (line 245-247)

5. Perhaps consider mentioning earlier in the METHODS, Patient cohort section, that patient demographic information (such as age, sex etc.) is available and point the reader to the appropriate tables.

Response: We added line 105 and 106 in the METHODS section with these pieces of information.

6. Regarding patient demographics, the supplemental table includes the exact date of birth and date of admission. This may be too identifying. The age (in years) at presentation may be sufficient, and the authors already state that patients presented/samples were collected in April 2020. Please consider if this type of (potentially identifying) information should be removed from the supplement.

Response: Thank you very much for highlighting this issue, we removed from S2 Table all potentially identifying data.

7. Gene names should be in italics (as an example, in the Introduction, line 59; but please check elsewhere).

Response: This has been fixed.

Reviewer #2: The authors re-evaluate the link between clonal hematopoiesis and COVID-19 severity by increasing the detection sensitivity of the defining somatic mutations (threshold 0.5%) in 24 patients with severe disease, 12 asymptomatic patients and 9 non-infected donors. The tested gene panel includes 80 genes, the coverage is 1350X, COSMIC and cBioportal are used to identify driver genes with a VAF that, in more than 80% of cases, was in the 0.5-2% range. The total number of detected mutations was similar in both group. Then, potential driver mutations and clonal complexity were analyzed, and these two parameters were higher in severe patients, without significant impact on clinical and biological parameters and disease outcome in this small cohort. Altogether, this study further argue for a limited impact of clonal hematopoiesis on severe COVID-19 outcome. The manuscript is clearly written and complements a series of previous analyses with various conclusions in small cohorts. A meta analysis will be needed for solid conclusions regarding the outcome. This report is a useful contribution to prepare this analysis.

In the introduction, the authors may stick to WHO 2022 classification of CHIP (Khoury et al, Leukemia 2022): CHIP refers to somatic mutations of myeloid malignancy-associated genes detected in the blood or bone marrow at a VAF ≥ 2% (≥4% for X-linked gene mutations in males) in individuals without a diagnosed hematologic disorder and without unexplained cytopenia.

Response: In the INTRODUCTION, line 51-55, we revisited the definition of CHIP according to the current WHO classification and cited the relevant publication, as suggested by the Reviewer.

The WHO classification also defines clonal cytopenia of undetermined significance (CCUS) as CHIP in the presence of one or more persistent cytopenias that are otherwise unexplained by hematologic or non-hematologic conditions and that do not meet diagnostic criteria for defined myeloid neoplasms. Did some of the patients included in this trial demonstrate a cytopenia before inclusion?

Response: None of the patients admitted to ICU had a positive clinical history for hematological disease and/or cytopenia. In the supplementary S2 Table, we report the results of the counts at admission to ICU of erythrocytes (RBC counts) and leucocytes (WBC counts). Counts are within the normal or normal-increased range for all patients. 

The introduction also focuses on the cardiovascular risk of CHIP but many other risks have now been identified, the latest being liver diseases (see B Ebert’s paper in Nature recently).

Response: We updated the INTRODUCTION (from line 67 to line 69) and cited the above paper within the REFERENCES.

Since they were included in 2020, do the authors have some information regarding the long-term outcome of the survivors?

Response: At one-year post-discharge from ICU, all patients included in the study were alive. We added a column reporting this piece of information in the S2 Table.

---

## [Decision Letter · Decision Letter 1]

18 Sep 2023

PONE-D-23-04485R1High-sensitivity Analysis of Clonal Hematopoiesis Reveals Increased Clonal Complexity of Potential-Driver Mutations in Severe COVID-19 PatientsPLOS ONE

Dear Dr. Ronchini,

Thank you for submitting your manuscript to PLOS ONE. After careful consideration, we feel that it has merit but does not fully meet PLOS ONE’s publication criteria as it currently stands. Therefore, we invite you to submit a revised version of the manuscript that addresses the points raised during the review process.

We look forward to receiving your revised manuscript.

Kind regards,

Antonio Solimando

Academic Editor

PLOS ONE

Journal Requirements:

Additional Editor Comments :

The author should follow reviewer 1 indication and acknowledge in the final version of the manuscript the following limitations:

Small Sample Size: The study uses a relatively small sample size, which can limit the generalizability of the results. Further studies with larger patient cohorts should be undertaken to validate the findings.

Cross-Sectional Data: The study presents data captured at a single point in time, without analyzing the VAF variation over time. This cross-sectional approach might not capture the dynamic nature of clonal hematopoiesis and its implications in disease progression. Future studies should consider a longitudinal approach to track VAF variations over time to understand the role of these mutations in disease trajectory.

Age-Dependent Variation: While the study acknowledges age-dependent variations, it seems the age groups compared have different sample sizes, which might affect the statistical power of the analysis. Implementing age-matched control groups would strengthen the study.

Clinical Significance of CH-PD: The study identified a higher percentage of CH-PD mutations in the COV-ICU group compared to the control group; however, the clinical significance of these mutations, in terms of their impact on patient outcomes, remains unclear. The manuscript could benefit from a deeper analysis examining the clinical relevance of these findings, potentially linking it to patient outcomes or specific clinical characteristics.

Lack of Long-Term Follow-Up: The study does not follow up on the potential development of more serious conditions like MDS/AML based on the identified mutations, missing an opportunity to explore the long-term implications of the identified CH-PDs.

Implementations that will be needed in future studies (if beyond the scope of this manuscript):

Longitudinal Study: Implement a longitudinal study design to better assess the variations of VAF over time and potentially capture more nuanced relationships between CH, age, and disease severity.

Expanded Genetic Analysis: Consider an expanded analysis involving a greater number of genes to potentially uncover more associations between genetic variations and disease outcomes.

Multivariate Analysis: Incorporate multivariate analyses to control for potential confounding factors and to robustly examine the association between the identified mutations and disease outcomes.

Questions to be Addressed

VAF Variation over Time: The authors should discuss if and how VAF variation over time (months/years)can influence the outcomes and the observed CH. This can potentially be a marker for disease progression and should be considered in future studies.

Relation to CHIP/CCUS ICUS: It would be beneficial for the authors to discuss whether the allele frequencies observed can be a mirror of CHIP/CCUS ICUS or frankly MDS/AML in some or other cases. An acknowledgment of this potential association and a suggestion to explore this in future research would enrich the discussion.

Reviewers' comments:

Reviewer's Responses to Questions

**Comments to the Author**

1. If the authors have adequately addressed your comments raised in a previous round of review and you feel that this manuscript is now acceptable for publication, you may indicate that here to bypass the “Comments to the Author” section, enter your conflict of interest statement in the “Confidential to Editor” section, and submit your "Accept" recommendation.

Reviewer #1: (No Response)

Reviewer #2: All comments have been addressed

2. Is the manuscript technically sound, and do the data support the conclusions?

Reviewer #1: Yes

Reviewer #2: Yes

3. Has the statistical analysis been performed appropriately and rigorously? 

Reviewer #1: Yes

Reviewer #2: Yes

4. Have the authors made all data underlying the findings in their manuscript fully available?

Reviewer #1: Yes

Reviewer #2: Yes

5. Is the manuscript presented in an intelligible fashion and written in standard English?

Reviewer #1: Yes

Reviewer #2: Yes

6. Review Comments to the Author

Reviewer #1: I thank the authors for considering and addressing the majority of my comments. I am satisfied with the overall changes and believe these have improved the manuscript.

I do ask the authors to check the numbering of references and to ensure this has not been disrupted. This came to my attention when following references 30-32, the criteria for CH and PD-CH variants. In the original submission, these correctly referred to studies by Bolton et al., Young et al. and Acuna-Hidalgo et al. If I am not mistaken, the numbering of references has shifted in this revised version, and now 30-32 correspond to Abelson et al. (prediction of AML risk in CH), Fabre et al. (study of CH in twins), and Bolton et al. I do not believe this is correct. This problem may be more extensive than mentioned. Please review carefully reference numbers in text versus the reference list to ensure matching. Thank you.

Reviewer #2: (No Response)

7. PLOS authors have the option to publish the peer review history of their article (what does this mean?). If published, this will include your full peer review and any attached files.

Reviewer #1: No

Reviewer #2: **Yes: **Eric Solary

---

## [Author Response · Author response to Decision Letter 1]

11 Oct 2023

We thank the Editor and the Reviewers for their comments. In the following pages are our point-by-point responses. 

Journal Requirements:

Response: we carefully reviewed our reference list and it appears complete and correct. By checking each journal website, none of the cited manuscript appears to be retracted.

Additional Editor Comments:

The author should follow reviewer 1 indication and acknowledge in the final version of the manuscript the following limitations:

Small Sample Size: The study uses a relatively small sample size, which can limit the generalizability of the results. Further studies with larger patient cohorts should be undertaken to validate the findings.

Response: We added a comment in the DISCUSSION section (line 298-301)

Cross-Sectional Data: The study presents data captured at a single point in time, without analyzing the VAF variation over time. This cross-sectional approach might not capture the dynamic nature of clonal hematopoiesis and its implications in disease progression. Future studies should consider a longitudinal approach to track VAF variations over time to understand the role of these mutations in disease trajectory.

Response: We added a comment in the DISCUSSION section (line 301-305)

Age-Dependent Variation: While the study acknowledges age-dependent variations, it seems the age groups compared have different sample sizes, which might affect the statistical power of the analysis. Implementing age-matched control groups would strengthen the study.

Response: One of the aims of our study was to analyze and compared samples from COV-ICU patients and controls that were collected during the same phase of the Covid-19 pandemic in the same regional area, in order to minimize the probability of infections from different viral variants, that clearly showed different microbiological characteristics and, more importantly, completely different clinical impacts on the infected individuals. Moreover, we wanted to analyze controls collected before the vaccination campaign, again, in order to avoid possible confounding effects on the hematopoietic cell system of our samples. In order to fulfill these requirements, for the controls, we took advantage of the study SOS-COV2 that included personnel working at our Institute and we analyzed all available samples collected from the oldest individuals of this cohort. 

Clinical Significance of CH-PD: The study identified a higher percentage of CH-PD mutations in the COV-ICU group compared to the control group; however, the clinical significance of these mutations, in terms of their impact on patient outcomes, remains unclear. The manuscript could benefit from a deeper analysis examining the clinical relevance of these findings, potentially linking it to patient outcomes or specific clinical characteristics.

Response: In the manuscript all available clinical data, patient outcome and follow-up at one year for the COV-ICU cohort are included (S2 Table). As shown in S4 Figure entitled: “Survival and clinical parameters in COV-ICU patients”, comparing COV-ICU patients with CH-PD mutations versus patients without CH-PD mutations, we did not identify statistically significant differences. This is acknowledged in the Results section (line 201-203). We added the reference to S2 Table and the no significance of data.

Lack of Long-Term Follow-Up: The study does not follow up on the potential development of more serious conditions like MDS/AML based on the identified mutations, missing an opportunity to explore the long-term implications of the identified CH-PDs.

Response: The absolute risk of hematopoietic malignancy development in persons with CH is low and individuals with CH progress to malignancy at a rate of about 0.5 to 1% per year. It has been reported that 4% of persons with CH develop hematopoietic malignancies in 8 years (Jaiswal et al., NEJM 2014). In the study from Abelson et al. (Nature 2018) individuals with CH developed myeloid neoplasms with a median follow-up of 7.6 years. Moreover, quantitatively the authors found that CH-PD mutations conferred around a 2-fold increase risk of developing AML per 5% increase in clone size. Moreover, CH parameters that increase the risk of leukemia transformation are linked to the presence of: clones with VAF>10%, more than 1 mutated gene, mutations in specific driver genes and specific aminoacid changes. Considering that: i) our cohort is small, ii) the COV-ICU patients had normal hematological counts and iii) in our COV-ICU patients the majority of clones have VAF<2%, we believe that the probability of scoring the development of myeloid neoplasms in the time frame of follow-up of 3 is extremely low. Moreover, although very interesting and important, this is beyond the scope of our study, which was the assessment of CH at high sensitivity in the two cohorts.

Implementations that will be needed in future studies (if beyond the scope of this manuscript):

Longitudinal Study: Implement a longitudinal study design to better assess the variations of VAF over time and potentially capture more nuanced relationships between CH, age, and disease severity.

Expanded Genetic Analysis: Consider an expanded analysis involving a greater number of genes to potentially uncover more associations between genetic variations and disease outcomes.

Multivariate Analysis: Incorporate multivariate analyses to control for potential confounding factors and to robustly examine the association between the identified mutations and disease outcomes.

Response: We agree with the editor that all these issues are very insightful. However, they would require the design and development of new specific trials and tools that are beyond the scope of this manuscript.

Questions to be Addressed

VAF Variation over Time: The authors should discuss if and how VAF variation over time (months/years)can influence the outcomes and the observed CH. This can potentially be a marker for disease progression and should be considered in future studies.

Response: We added a comment in the DISCUSSION section (line 236-250)

Relation to CHIP/CCUS ICUS: It would be beneficial for the authors to discuss whether the allele frequencies observed can be a mirror of CHIP/CCUS ICUS or frankly MDS/AML in some or other cases. An acknowledgment of this potential association and a suggestion to explore this in future research would enrich the discussion.

Response: Most of the genomic studies on myeloid neoplasms are based on whole exome or large gene panels sequencing analysis and usually, consistent with normal clinical practice, use VAF threshold of 1% for hotspot mutations and of 5% for variants of unknown significance. Moreover, in frank myeloid neoplasms usually important clonal expansions has already taken place. It is, therefore, not surprising that most of the mutations in common with our study display higher VAF in dataset of myeloid neoplasms or CHIP compared to the one we scored. Nonetheless, we found similar VAF in a few patients with myeloid neoplasms and we added a comment addressing this issue in the DISCUSSION section (line 256-268).

Reviewer #1: I thank the authors for considering and addressing the majority of my comments. I am satisfied with the overall changes and believe these have improved the manuscript.

I do ask the authors to check the numbering of references and to ensure this has not been disrupted. This came to my attention when following references 30-32, the criteria for CH and PD-CH variants. In the original submission, these correctly referred to studies by Bolton et al., Young et al. and Acuna-Hidalgo et al. If I am not mistaken, the numbering of references has shifted in this revised version, and now 30-32 correspond to Abelson et al. (prediction of AML risk in CH), Fabre et al. (study of CH in twins), and Bolton et al. I do not believe this is correct. This problem may be more extensive than mentioned. Please review carefully reference numbers in text versus the reference list to ensure matching. Thank you.

Response: The numbering of references changed from the original manuscript because, as mentioned in the Rebuttal letter for the first round of revision, we added new references, according to the suggestions by the Reviewers. We carefully checked the numbering reported in the revised manuscript and it is correct. Indeed, for annotation of CH variants as potential drivers, we used the definitions and the parameters reported in Bolton et al., Abelson et al. and Fabre et al. In particular, we used as reference the Supplemental Table 2 published in Fabre et al. plus all parameters described in the other manuscripts. To the best of our knowledge, there is no a formal definition of CH-Driver mutations in Young et al. or Acuna-Hidalgo et al.

---

## [Editor Report · Decision Letter 2]

7 Nov 2023

PONE-D-23-04485R2High-sensitivity Analysis of Clonal Hematopoiesis Reveals Increased Clonal Complexity of Potential-Driver Mutations in Severe COVID-19 PatientsPLOS ONE

Dear Dr. Ronchini,

Thank you for submitting your manuscript to PLOS ONE. After careful consideration, we feel that it has merit but does not fully meet PLOS ONE’s publication criteria as it currently stands. Therefore, we invite you to submit a revised version of the manuscript that addresses the points raised during the review process.

We look forward to receiving your revised manuscript.

Kind regards,

Antonio Solimando

Academic Editor

PLOS ONE

Journal Requirements:

Additional Editor Comments:

Overall, the authors have made efforts to address the reviewers' comments and justify any limitations or decisions made in the study, while also updating the reference list and clarifying any confusion around it. They are open about the limitations of their study and have incorporated suggestions from the reviewers into the revised manuscript where possible. They also highlight areas that could be explored in future research but which were beyond the scope of their current study. Indeed, the authors' responses in the rebuttal letter generally address the concerns raised by the reviewers. However, there are always areas for improvement, both in the manuscript and in the way the authors communicate their revisions to the reviewers and the readers. Here are some suggestions:

Clarification on Reference Numbering:

The response about the reference numbering was somewhat unclear. The authors should make sure that the reference numbers in the text match the updated reference list. They should explicitly confirm that they have double-checked the in-text citations against the updated reference list to ensure all references are accurately represented.

Explicit Acknowledgement of Limitations:

While the authors have acknowledged the limitations pointed out by the reviewers in the discussion section, it may be beneficial to include a succinct summary of these limitations in the abstract or conclusions to ensure readers are immediately aware of the study's scope and constraints.

Methodological Rigor and Follow-Up:

Although the authors mention that long-term follow-up and expanded studies are beyond the scope of the current study, they could outline a more detailed plan for future research. This might include potential study designs, collaborations, or funding opportunities they plan to seek in order to address these limitations.

Detailed Justification for Methodological Choices:

For the controls chosen from the SOS-COV2 study, the authors should provide a more detailed rationale for why these specific samples were selected, beyond the timing of collection and the pre-vaccination status, such as demographic similarities or exposure risks that match the study group.

Statistical Considerations:

The authors should consider discussing any additional statistical tests or models that could be applied to their data in future studies to account for confounding variables not addressed in this study, such as multivariate regression analyses or propensity score matching.

Potential Implications and Applications:

A deeper exploration of how the identified mutations could be monitored or used in clinical practice might add value to the discussion, even if this application is not immediately feasible.

Accessibility of Data and Materials:

Ensuring that all supplementary materials and methods are easily accessible to readers and researchers for reproducibility purposes is important. If the authors have not already done so, they might consider including a statement confirming the availability of such resources or outlining how they can be accessed.

---

## [Author Response · Author response to Decision Letter 2]

10 Nov 2023

In the following pages are our point-by-point responses. 

Journal Requirements:

Response: we double-checked the in-text citations against the updated reference list and ensured that all references are accurately represented. By checking each journal website, none of the cited manuscript appears to be retracted.

Additional Editor Comments:

Overall, the authors have made efforts to address the reviewers' comments and justify any limitations or decisions made in the study, while also updating the reference list and clarifying any confusion around it. They are open about the limitations of their study and have incorporated suggestions from the reviewers into the revised manuscript where possible. They also highlight areas that could be explored in future research but which were beyond the scope of their current study. Indeed, the authors' responses in the rebuttal letter generally address the concerns raised by the reviewers. However, there are always areas for improvement, both in the manuscript and in the way the authors communicate their revisions to the reviewers and the readers. Here are some suggestions:

Clarification on Reference Numbering:

The response about the reference numbering was somewhat unclear. The authors should make sure that the reference numbers in the text match the updated reference list. They should explicitly confirm that they have double-checked the in-text citations against the updated reference list to ensure all references are accurately represented.

Response: we confirm that we double-checked the in-text citations against the updated reference list and ensured that all references are accurately represented.

Explicit Acknowledgement of Limitations:

While the authors have acknowledged the limitations pointed out by the reviewers in the discussion section, it may be beneficial to include a succinct summary of these limitations in the abstract or conclusions to ensure readers are immediately aware of the study's scope and constraints.

Response: we acknowledged the limitations of our study in the abstract (line 49-53).

Methodological Rigor and Follow-Up:

Although the authors mention that long-term follow-up and expanded studies are beyond the scope of the current study, they could outline a more detailed plan for future research. This might include potential study designs, collaborations, or funding opportunities they plan to seek in order to address these limitations.

Response: A possible strategy would be to retrospectively retrieve DNA from PB that have been collected from COVID19 patients treated in ICU and, then, longitudinally at different follow-up post-discharge, in order to perform our high sensitivity CH analysis. However, we really believe this is beyond the scope of the current study.

Detailed Justification for Methodological Choices:

For the controls chosen from the SOS-COV2 study, the authors should provide a more detailed rationale for why these specific samples were selected, beyond the timing of collection and the pre-vaccination status, such as demographic similarities or exposure risks that match the study group.

Response: we added these details in the Materials and Methods section (line 110-118).

Statistical Considerations:

The authors should consider discussing any additional statistical tests or models that could be applied to their data in future studies to account for confounding variables not addressed in this study, such as multivariate regression analyses or propensity score matching.

Response: Increasing the number of patients analyzed, one could apply machine learning models to discriminate between ICU patients and Controls by integrating multivariate clinical features with our genomic features. However, we believe it is premature to suggest any additional specific type of statistical tests or models without additional available information.

Potential Implications and Applications:

A deeper exploration of how the identified mutations could be monitored or used in clinical practice might add value to the discussion, even if this application is not immediately feasible.

Response: We applied a high sensitivity NGS technology which requires extensive experimental and bioinformatic workflows, is expensive, and is not easily implementable in clinical practice. To our knowledge, up to day, the only alternative available technique with similar sensitivity is digital PCR, not used in standard clinical practice. Moreover, how to counsel people with CH or monitor them prospectively is an area of uncertainty and a quite challenging task at the moment. Indeed, there are no clinical indications or guidelines on how to manage the presence of CH mutations nor there are established interventions to try to eradicate CH clones. We believe it is really too early to include this information in our paper, especially, considering that we mainly detected low frequency mutations, which require high sensitivity techniques to be monitored.

Accessibility of Data and Materials:

Ensuring that all supplementary materials and methods are easily accessible to readers and researchers for reproducibility purposes is important. If the authors have not already done so, they might consider including a statement confirming the availability of such resources or outlining how they can be accessed.

Response: we added a Data Availability Statement (line 322-324).

---

## [Editor Report · Decision Letter 3]

28 Nov 2023

High-sensitivity Analysis of Clonal Hematopoiesis Reveals Increased Clonal Complexity of Potential-Driver Mutations in Severe COVID-19 Patients

PONE-D-23-04485R3

Dear Dr. Ronchini,

We’re pleased to inform you that your manuscript has been judged scientifically suitable for publication and will be formally accepted for publication once it meets all outstanding technical requirements.

Kind regards,

Antonio Solimando

Academic Editor

PLOS ONE
---

## [Editor Report · Acceptance letter]

28 Dec 2023

PONE-D-23-04485R3 

PLOS ONE

Dear Dr. Ronchini, 

I'm pleased to inform you that your manuscript has been deemed suitable for publication in PLOS ONE. Congratulations! Your manuscript is now being handed over to our production team.

Kind regards, 

on behalf of

Dr. Antonio Solimando 

Academic Editor

PLOS ONE